# Extraction Methods Determine the Quality of Soil Microbiota Acquisition

**DOI:** 10.3390/microorganisms12020403

**Published:** 2024-02-17

**Authors:** Zhuoxin Liu, Chi Zhang, Jiejia Ma, Qianze Peng, Xiaohua Du, Shu’e Sun, Ju’e Cheng, Weiye Peng, Lijie Chen, Zepei Gu, Weixing Zhang, Pin Su, Deyong Zhang

**Affiliations:** 1Longping Branch, College of Biology, Hunan University, Changsha 410082, China; xin99@hnu.edu.cn (Z.L.); czh113@hnu.edu.cn (C.Z.); mjj1999@hnu.edu.cn (J.M.); 18153785730@163.com (L.C.); 2State Key Laboratory of Hybrid Rice, Institute of Plant Protection, Hunan Academy of Agricultural Sciences, Changsha 410125, China; pengqianze@hainanu.edu.cn (Q.P.); xiaohuadu@126.com (X.D.); sse692010@126.com (S.S.); chengxiaozhb@126.com (J.C.); 15580076854@163.com (W.P.); gzp2537342238@163.com (Z.G.); 17793687462@163.com (W.Z.); 3College of Tropical Crops, Hainan University, Haikou 570228, China; 4National Center of Technology Innovation for Saline-Alkali Tolerant Rice in Sanya City, Sanya 572024, China

**Keywords:** extraction microbiota, microbiota diversity, sonication, oscillation and processing times

## Abstract

The soil microbiome plays a key role in plant health. Native soil microbiome inoculation, metagenomic profiling, and high-throughput cultivation require efficient microbe extraction. Sonication and oscillation are the most common methods used to extract soil microbiomes. However, the extraction efficiency of these methods has not been investigated in full. In this study, we compared the culturable microbe numbers, community structures, and alpha diversities among the different methods, including sonication, oscillation, and centrifugation, and their processing times. The study results showed that sonication significantly increases the culturable colony number compared with oscillation and centrifugation. Furthermore, the sonication strategy was found to be the main factor influencing extraction efficiency, but increased sonication time can aid in recovery from this impact. Finally, the extraction processing times were found to have a significant negative relationship with α-diversity among the extracted microbiota. In conclusion, sonication is the main factor for enriching in situ microbiota, and increased extraction time significantly decreases the α-diversity of the extracted microbiota. The results of this study provide insights into the isolation and utilization of different microorganism sources.

## 1. Introduction

Microbes [1] perform a variety of vital functions that are essential for healthy ecosystems, ranging from nutrient recycling and antibiotic production, to waste decomposition. Many in situ extracted microorganisms are used to improve human health [2], control environmental pollution [3], and enhance agricultural production [4]. One of the most useful microbiome transplants is fecal microbiota transplantation (FMT) [5]. FMT has been used to treat a variety of human diseases, such as infectious diseases [6,7], inflammatory bowel diseases [8,9,10,11,12], oncological diseases [13], hematological diseases [14,15], and neurodegenerative diseases [16]. Additionally, microbes make an enormous contribution to environmental governance and can also assist in the elimination of pollutants from hyperthermal, acidic, hypersaline, or basic industrial waste [17,18]. Microbial biotechnology offers sustainable routes to plastic production and waste management [3]. Microorganisms are an important element in modeling sustainable agriculture [19,20] and are crucial in maintaining plants’ growth, development, and yield [21,22,23,24]. Plant-associated microbial communities play a pivotal role in plant nutrient acquisition and nitrogen and carbon cycling and also aid in helping crops tolerate biotic and abiotic stresses [24].

Most microbial communities in nature exist in complex, dynamic consortia. Microbial diversity contributes to many ecosystem services, such as water storage, carbon sequestration, the maintenance of soil structure, plant productivity, and pest and disease suppression [24]. Most land crops are closely associated with a complex diversity of microorganisms. These crop-related microorganisms regulate crop growth and improve crop yield and quality by direct (promoting nutrient acquisition and regulating plant hormone levels) or indirect (inducing systemic resistance and biosynthesis) mechanisms [25]. For example, nitrogen-fixing bacteria, potassium-dissolving bacteria, and phosphorus-dissolving microorganisms can enhance plant nutrient absorption and productivity by promoting nutrient circulation and increasing soil fertility. Rhizobium nitrogen fixation helps legumes obtain nitrogen and improve their yield [21]. To acidify biological cells and the surrounding environment, potassium- and phosphorus-solubilizing microorganisms release specific organic substances (such as oxalic acid, glucose, and malic acid). This causes potassium or phosphate, which are originally insoluble minerals, to be released into the soil, thereby increasing the availability of effective nutrients in the soil, promoting plant growth, and improving the yield and quality of crops. Microbes can also enhance crop quality by regulating plant hormones, such as indoleacetic acid, ethylene, auxin, and gibberellin. For example, the diversity of microorganisms enhances strawberry biomass through the production of indoleacetic acid and the adjustment of abscisic acid levels, which increases the total sugar concentration of the fruit and improves its sweetness [25]. In addition, microbial diversity plays an important role in ecosystem functions. Biodiversity enhances ecosystem stability and productivity [26].

To maintain microbial diversity, the use of in situ microbiome engineering methods allows for the manipulation and study of microbial communities in their native context and minimizes the overall impact on the community. Recently, it has been demonstrated that the plant microbiome can be modified through the transplantation of the microbiota, with this method yielding substantial results for the control of plant diseases [1]. Therefore, microbiota extraction methods are an important element in isolated and transplanted microbes.

Microbiota extraction methods encompass sonication, oscillation, and centrifugal processes and are used to enrich in situ microbiota [24,27,28]. However, few studies have focused on the factors that impact the extraction of microbiota. In this study, we first tested the influence of centrifugation, oscillation, and sonication processes on the effectiveness of the culturable bacterial numbers, community structure, and diversity of the extracted soil microbiota. We found that sonication showed the most significant effect on the extracted microbiota. Afterward, methods with different sonication strategies and time periods were formulated to explore how sonication affects extraction efficiency. At the same time, we analyzed the correlation between processing time and diversity in an attempt to better explain the previous results. In conclusion, our study promotes a further understanding of the impact of different extraction methods on microbiota, which can aid in providing better insights into the isolation and utilization of different microorganism sources.

## 2. Materials and Methods

### 2.1. Soil Sampling

The soil used in our experiments was collected from a rice field in Taojiang County, Yiyang city, Hunan province, China (28°38′09″ N, 112°0′57″ E). The top 3~20 cm of the soil was collected and sieved (using a 3 mm sieve) to remove rocks and other debris. The soil was air-dried overnight at room temperature until use.

### 2.2. Design of the Soil Microbiota Extraction Methods

In order to compare the effect sizes of centrifugation, oscillation, and sonication on soil microbiota extraction, we designed four groups of methods (CK, CF, UT1, and LT; Table 1). For each method, 10 g of prepared soil was mixed with 50 mL of sterile water in a conical bottle. To extract the microbiota, the following procedures were utilized: oscillation for 30 min at 200 rpm (CK); oscillation for 3 h at 200 rpm (LT); and sonication for 2 min at a frequency of 30 kHz after oscillation for 30 min at 200 rpm (UT1). Afterward, the suspensions were incubated for 15 min at room temperature to precipitate the soil. The supernatant was then the extracted microbiota. In addition, the CK-method-extracted microbiota was subjected to centrifugation (600 rpm, 1 min, 4 °C) as a CF method. The extracted microbiota was stored at −80 °C before further analysis.

### 2.3. Investigation of the Effect Size of Sonication Time and Strategy among the Extracted Microbiota

Based on the above analysis, we found that sonication was the main factor involved in microbiota extraction (Appendix A). Furthermore, to compare the effect size of sonication time and strategy, we designed two methods to allow for comparison with the previous method, UT1. Based on the method UT1, oscillation time was divided into two equal parts (15 min), and we employed the sonication treatment in the middle of this process as a new sonication strategy. In detail, this involved oscillation for 15 min at 200 rpm, with sonication employed for 2 min (as UT2) or 6 min (as UT3) at a frequency of 30 kHz, and then, we continued oscillation for 15 min at 200 rpm. The extracted microbiota was stored at −80 °C until further analysis.

### 2.4. Plate Counting

To accurately quantify the culturable bacterial populations within the extracted microbiota, we employed the standard plate counting method to enumerate viable bacteria [29]. The extracted microbiota was diluted 10^6^ times using sterile water through a ten-fold series continuous dilution method, and then, 70 μL of diluted microbiota was spread on nutrient broth agar (NA) medium. The plate was incubated overnight at 28 °C before the number of colony-forming units (CFUs) was calculated. Each extraction method was repeated 18 times. Plotted and unpaired one-way analysis of variance (ANOVA) with Tukey’s test was conducted using GraphPad Prism 8 software (San Diego, CA, USA).

### 2.5. 16s rRNA Amplicon Sequencing

The extracted microbiota was analyzed to determine the microbiota diversity through the use of 16S rRNA sequencing (three replicates per method, with each replicate mixed with three extracted microbiota). Total DNA was extracted using a MagPure Soil DNA LQ Kit (Magen, Shanghai, China) based on the manufacturer’s instructions. A NanoDrop ND-1000 spectrophotometer (Thermo Fisher Scientific, Waltham, MA, USA) and agarose gel electrophoresis were used to test the quality and quantity of DNA. The extracted DNA was diluted to a concentration of 1 ng/μL and stored at −20 °C for further analysis. PCR amplification of the bacterial 16S rRNA gene fragments (V3–V4 region) was performed using Takara Ex Taq (Takara, Beijing, China) and the barcoded primers 343F (5′-TACGGRAGGCAGCAG-3′) and 798R (5′-AGGGTATCTAATCCT-3′). Amplicons were visualized using agarose gel electrophoresis and purified using Agencourt AMPure XP beads (Beckman Coulter, Pasadena, CA, USA) twice. After purification, the DNA was quantified using a Qubit dsDNA assay kit (Yeasen, Shanghai, China). Equal amounts of purified DNA were pooled for sequencing on the NovaSeq 6000 platform (Illumina Inc., San Diego, CA, USA) at Shanghai OEbiotech (Shanghai, China).

### 2.6. Data Analysis

The 16S rRNA gene fragment sequences were processed using QIIME2 v.2021.4 [30]. Paired-end reads were detected, and the adapters were removed using vsearch v.2.26.1 [31]. After trimming, paired-end reads were filtered for low-quality sequences, denoised, merged, and clustered using DADA2 v.2020.2.0 [32]. Amplicon sequence variants (ASVs) and a feature table were generated using QIIME2. All ASVs were annotated using the Silva v138.1 reference databases [33]. Principal coordinate analysis (PCoA) based on Bray–Curtis distances was performed using the R package vegan [34]. Diversity and differential abundance analyses were performed using STAMP v.2.1.3 software [35].

## 3. Results

### 3.1. Sonication Increased Culturable Bacteria Diversity

To determine the culturable bacteria diversity of the extracted microbiota from three different extraction methods, we counted the bacterial numbers using plate counting (Figure 1 and Appendix A). The results showed that the culturable bacterial numbers extracted using the UT1 (sonication) method were significantly higher than those extracted using the CK, CF (centrifugation), and LT (oscillation) methods (Figure 1A, *p* < 0.05, unpaired one-way ANOVA with Tukey’s test). The culturable bacterial morphology extracted using the UT1 method showed more diversity than that extracted using the CK, CF, and LT methods (Figure 1B and Appendix A). However, bacterial numbers extracted using the CK, CF, and LT methods showed no significant difference (Figure 1A, *p* > 0.05) and similar diversity (Figure 1B, Appendix A, and Appendix A, respectively). These results indicate that sonication was the main factor involved in increasing culturable bacteria diversity.

### 3.2. Ultrasonication Was the Main Factor Impacting the Composition and Diversity of the Extracted Microbiota

To further investigate the effect size of microbiota diversity variation between the three extraction methods, we analyzed their microbiome using 16s rRNA sequencing. The principal coordinate analysis (PCoA) performed showed that the UT1-, LT-, and CK-extracted microbiome was significantly separated into three parts (*p* < 0.001, PERMANOVA by adonis, Figure 2B). And the first- and second-principal-coordinated axes (PCo1 and PCo2) explained 50.19% and 15.92% of the variation, respectively (Figure 2B). Comparing the microbiome obtained with CK, we found that the UT1, LT, and CF methods contributed 67.37%, 58.42%, and 29.43% to the effect size for microbiome variation (Figure 2C). To further investigate the compositional variation among the three extraction methods, we analyzed their taxonomy at the phylum and order levels (Figure 2A and Appendix A).

We found that Proteobacteria, Acidobacteria, Chloroflexi, Actinobacteria, and Nitrospirae were the top five enriched phyla among the CK-, UT1-, CF-, and LT-extracted microbiota (Appendix A). And the extraction method UT1 was found to be able to significantly enrich Proteobacteria and Actinobacteria compared with the CK method (*p* < 0.01, Appendix A). Acidobacteriales, Nitrospirales, Rhodospirillales, Burkholderiales, and Rhizobiales were the top five enriched orders among the CK-, UT1-, CF-, and LT-extracted microbiota (Figure 2). Compared with the CK method, the CF-, LT-, and UT1-extracted microbiota were significantly enriched in 1, 2, and 4 orders among the top 20 enriched orders, respectively (*p* <0.05, Figure 2D–F). The relative abundance of Chlorobiales in the microbiota extracted using the CF method was more significantly enriched than those extracted using the CK method (*p* < 0.05, Figure 2D). And the relative abundance of Sphingomonadales and Burkholderiales in microbiota extracted using the LT method was significantly higher than that in the microbiota extracted using the CK method (*p* < 0.05, Figure 2E). In addition, compared with the CK method, the microbiota extracted using the UT1 method significantly increased the relative abundance of Sphingomonadales, Neisseriales, Burkholderialesm, and Xanthomonadales (*p* < 0.05, Figure 2F). These results indicate that sonication significantly enriched the microbiota.

### 3.3. Sonication Time and Strategy Impact Culturable Bacteria Diversity

Many microbiota extraction methods involve sonication before oscillation and different sonication treatment times [36]. However, how these factors impact the extracted microbiota diversity has been scarcely investigated. Therefore, we designed a method of sonication treatment before oscillation, i.e., UT2, and a method of increased ultrasonic treatment time, i.e., UT3. Compared with UT1, culturable bacterial number and diversity gained through the use of the UT2 method significantly decreased (*p* < 0.05, Figure 3 and Appendix A). Interestingly, culturable bacterial number and diversity gained through the use of method UT3 (the increased sonication time) significantly increased compared to the UT2 method (*p* < 0.05, Figure 3A and Appendix A), and no significant difference compared to the UT1 method was observed (*p* > 0.05, Figure 3A and Appendix A). These results show that the different sonication strategies determine the decrease in culturable bacteria diversity, and that increased sonication time can aid in recovery from this impact.

### 3.4. The Sonication Strategy Impacts Microbiota Composition during Extraction 

In order to investigate the effect size of microbiome variation between the sonication time and strategy, we analyzed their extracted microbiota composition through 16s rRNA sequencing. The principal coordinate analysis results showed that the extracted microbiotas using the UT1 and UT2 methods were significantly separated into two clusters (*p* < 0.001, PERMANOVA by adonis, Figure 4B). And the first- and second-principal-coordinated axes (PCo1 and PCo2) explained 61.37% and 13.26% of the variation, respectively (Figure 4B). Compared with the method UT2, via which the microbiota extracted involved a different sonication strategy, UT1 showed a 68.88% effect size for microbiome variation. However, the microbiota extracted via the increased-sonication-time method UT3 simply showed a 26.26% effect size (Figure 4C). Through further analysis of the taxonomic diversity among the microbiota extracted via the three methods, we found that Proteobacteria, Acidobacteria, Chloroflexi, Actinobacteria, and Nitrospirae were the top five enriched phyla among the microbiota extracted via the three methods (Appendix A). Actinobacteria from the UT2-extracted microbiota significantly decreased compared to those from the UT1-extracted microbiota (*p* < 0.05, Appendix A). However, the UT3-extracted microbiota showed no significant difference compared to the UT2-extracted microbiota at the phylum level (*p* > 0.05, Appendix A).

Next, we analyzed the composition of the extracted microbiota at the order level. We found that Acidobacteriales, Nitrospirales, Rhizobiales, Rhodospirillales, and Nitrosomonadales were the top five enriched orders among the microbiota extracted via the three methods (Figure 4A). Difference analysis showed that the relative abundance of the extracted microbiota using the UT2 and UT3 methods significantly decreased by 6 and 3 orders compared to the UT1 method among the top 20 enriched orders, respectively (Figure 4D). In detail, the relative abundance of Sphingomonadales, Chlorobiales, Xanthomonadales, Burkholderiales, Sphingobacteriales, and Desulfuromonadales extracted using the UT2 method significantly decreased compared to those extracted using UT1 (*p* < 0.05, Figure 4D). But compared with the UT1 method, the relative abundance of Sphingomonadales, Sphingobacteriales, and Chlorobiales among the microbiota extracted via the UT3 method significantly decreased (*p* < 0.05, Figure 4E). These results showed that ultrasonic strategy was a main effect factor for extracted microbiota variation, and an increase in ultrasonic time led to a difficult recovery of extracted microbiota diversity.

### 3.5. The Processing Time Significantly Decreased Extracted Microbiota α-Diversity

Additionally, we found that the culturable bacterial number and diversity gained through the use of the longer-processing-time method LT decreased (Figure 1A and Appendix A). Therefore, we hypothesized that the processing time may impact the extracted microbiota diversity. To investigate the correlation between processing time and microbiota diversity, we analyzed the Shannon index (α-diversity) across all methods used for microbiota extraction. The results showed that the method with a shorter processing time, CK, showed the highest Shannon index, and the method with the longest processing time, LT, showed a significantly lower Shannon index (Figure 5A). Interestingly, the method with a shorter processing time but with a change in the sonication strategy, UT2, also significantly decreased the Shannon index (Figure 5A); however, the increased-sonication-time method UT3 increased the Shannon index. These results are consistent with previous results that showed that the ultrasonic strategy was the main factor affecting the extracted microbiota diversity. Furthermore, we calculated all methods’ processing times and their correlation with the Shannon index. We found that the processing time showed a significantly negative relationship with the Shannon index of the extracted microbiota (*p* < 0.05, Figure 5B). These results lead to the conclusion that the processing time and ultrasonic strategy are the primary factors influencing the microbiota extraction process.

## 4. Discussion

Over an extended period of time, the microbiome has been consistently acknowledged as a pivotal constituent of plant ecosystems. Within the realm of microbiota research, microbial extraction is an indispensable procedural facet. Nevertheless, the contemporary literature is relatively scant regarding methodologies for microbiome extraction. Within the scope of this investigation, diverse approaches to extracting soil microbial communities were methodically designed. Specifically, we first tested the effective influence of centrifugation, oscillation, and sonication on the culturable bacterial numbers, community structure, and diversity of the extracted soil microbiota. Sonication showed the most significant effect on the extracted microbiota. Afterward, methods with different sonication strategies and times were designed to explore how sonication affects extraction efficiency, and the results showed that the sonication strategy was the main factor influencing extraction efficiency, and increasing the sonication time can aid in recovery from this impact. Finally, we analyzed the α-diversity among the microbiota extracted via all methods and their correlation with the processing times. The α-diversity among the extracted microbiota showed a significant negative relationship with the processing time. For each extraction factor, firstly, in terms of oscillation, we found that oscillation had not been used in the extraction process of Actinobacteria in past studies [37,38], and in this study, it was also confirmed that the length of the oscillation time did not affect Actinobacteria. Conversely, within the realm of rhizosphere Acidobacteria, varying agitation times, spanning from 0 min [37] to 1 h [39], were found to engender a decrement in the relative abundance of Acidobacteria. This observation aligns with our findings, where LT treatment led to a significant reduction in Acidobacteria abundance compared to CK treatment (*p* < 0.05, Figure 4). The decrease in microbial community α-diversity caused by increasing oscillation time, which may be related to the length of processing time, will be discussed further ahead.

Historically, research has harnessed ultrasonication technology to recover microorganisms from wood surfaces, demonstrating superior yields compared to grinding, without inducing significant microbial mortality [40]. Notably, ultrasonication has been employed in the medical domain to collect bacterial samples from biofilms adhering to prosthetic surfaces [41]. Furthermore, in the water treatment industry, ultrasonication systems have exhibited superior microbial control capabilities [42]. These instances underscore the pivotal role of ultrasonic technology in microbiome extraction. It is noteworthy that low-intensity ultrasound alone may not effectively kill bacteria. Instead, the stable cavitation of ultrasound facilitates alterations in bacterial cell membranes, influencing bacterial adhesion and growth [43]. This phenomenon likely renders microorganisms more amenable to extraction from a diverse array of materials, a premise that aligns with the outcomes of this study where ultrasonication led to an increased cultivable bacterial count in extractions (Figure 1), emerging as a primary determinant of microbiome extraction efficiency. Nevertheless, it should be noted that certain studies have employed post-ultrasonication agitation as a strategy to enhance the culturing efficiency of bacteria isolated from the ultrasonication process. Contrarily, the findings presented in this study, as illustrated in Appendix A, suggest that this methodology might lead to a reduction in the diversity of the extracted microbial population. This observed effect could potentially be ascribed to the reattachment of bacteria to the substrates during the incubation phase, which necessitates further investigation.

Finally, we found that the extracted microbiota α-diversity was significantly reduced with the extension of the processing time (Figure 5). Studies have shown that microbial environmental fluctuations can induce responses from microbial communities, populations, and individuals on a time scale [44]. In a study on the effects of microbial soil amendments on the bacterial microbiome of strawberry roots, it was found that the longer the use of the amendments, the lower the α-diversity of the strawberry root microbial community [36], which is consistent with our findings. Soil pores are usually filled with different proportions of air and nutrients. Due to the prolonged processing time involved in the extraction process, nutrients and oxygen are lost, and microorganisms compete for scarce resources [45], resulting in changes in microbial communities and affecting microbial activity. There is a vertical habitat heterogeneity in space and time between nutrients and oxygen in lakes, resulting in large differences in bacterial community structure [46,47], which is an excellent example. This may explain the significant reduction in the α-diversity of the extracted flora in this study as a result of the increased processing time, but we expect that changes in microbial diversity in response to different extraction methods are likely dependent on more factors. In conclusion, this study offers crucial insights into microbial community dynamics and the efficiency of various extraction techniques. Researchers can refine and optimize extraction protocols through the quantity and diversity of microbial communities.

## Figures and Tables

**Figure 1 microorganisms-12-00403-f001:**
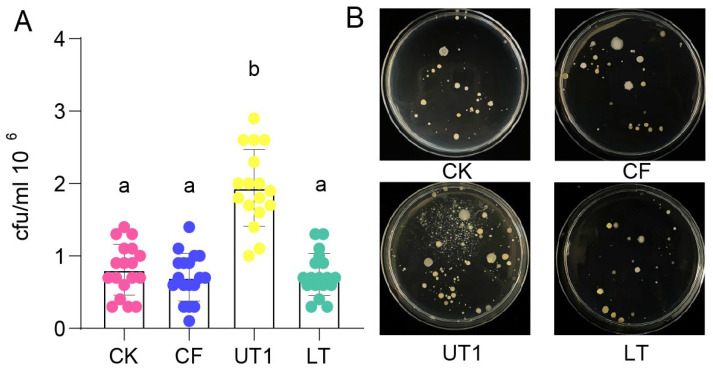
Effect of different extraction methods on the number of culturable bacteria. (**A**) CFU value of the soil bacterial suspension extracted using each extraction method. Letters indicated by one-way ANOVA and Tukey’s HSD; the same letters indicate no significant difference, while different letters indicate a significant difference (*p* < 0.0001). The different colors represent the different treatment groups: rose red, blue, yellow, and green represent CK, CF, UT1, and LT, respectively. The scatter plot represents the sample data, and bars represent the mean ± SD. (**B**) Colony phenotypes of soil bacteria culturable using each extraction method. Eighteen repeats of each extraction method were carried out. The groups are abbreviated as follows: control method, CK; centrifugation method, CF; sonication method, UT1; oscillation method, LT.

**Figure 2 microorganisms-12-00403-f002:**
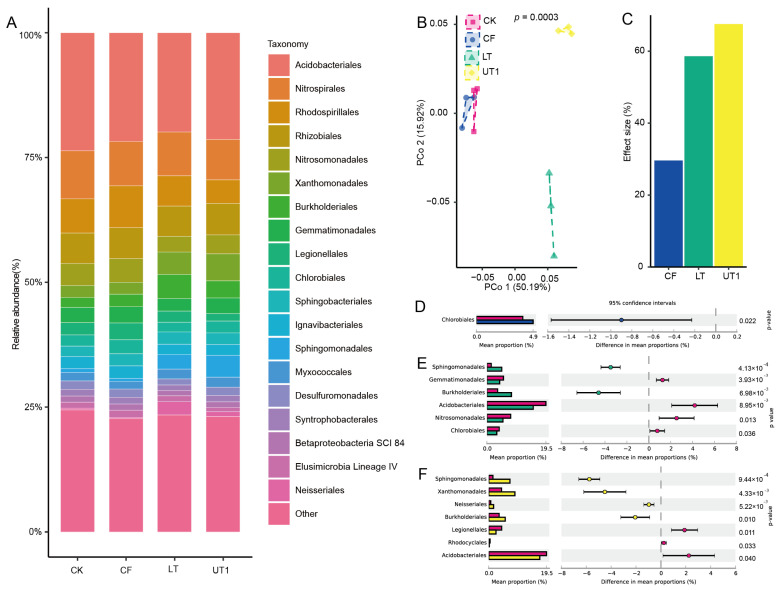
Effects of different extraction factors on bacterial community diversity and composition. (**A**) Relative abundance of bacterial communities extracted using the CK, CF, UT1, and LT methods at the order level. (**B**) The unconstrained PCoA based on Bray–Curtis distance showed bacterial community aggregation under different extraction methods (*p* < 0.001; *p*-value calculated using PERMANOVA). (**C**) The extracted bacterial community using each method was compared to the CK method during PCoA followed by PERMANOVA, and the effect size of the extracted method was plotted in increasing order. Differences analysis of the top 10 orders among the CK- and CF-extracted microbiota (**D**), the CK- and UT1-extracted microbiota (**E**), and the CK- and LT-extracted microbiota (**F**). The histogram shows the difference in bacterial abundance between the two groups. The dot and bar plots show the percentage of bacteria between the two groups in each sample. The difference in proportions between groups is shown with 95% confidence intervals. Only *p* < 0.05 (Welch’s *t*-test) is shown. Each method was repeated three times. The abbreviations and different colors are the same as those shown in Figure 1.

**Figure 3 microorganisms-12-00403-f003:**
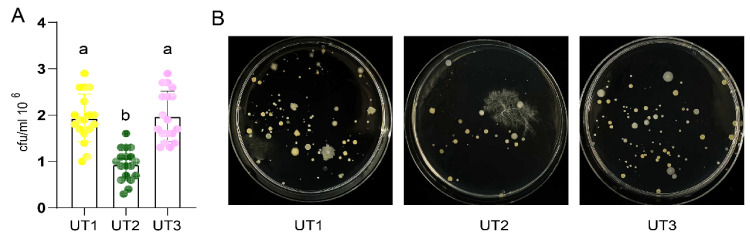
Effects of the different sonication methods on the bacterial culture. (**A**) CFU value of the soil bacterial suspension extracted from each treatment. Letters indicated by one-way ANOVA and Tukey’s HSD; the same letters indicate no significant difference, while different letters indicate a significant difference (*p* < 0.0001). Different colors represent the different treatment groups: yellow, dark green, and pink represent UT1, UT2, and UT3 treatments, respectively. The scatter plot represents the sample data, and bars represent the mean ± SD. (**B**) Colony phenotypes of soil bacteria culturable in each treatment. Eighteen repeats per extraction method were carried out. The groups are abbreviated as the following: sonication method, UT1; sonication method using the new strategy, UT2; increased sonication time method with the new strategy, UT3.

**Figure 4 microorganisms-12-00403-f004:**
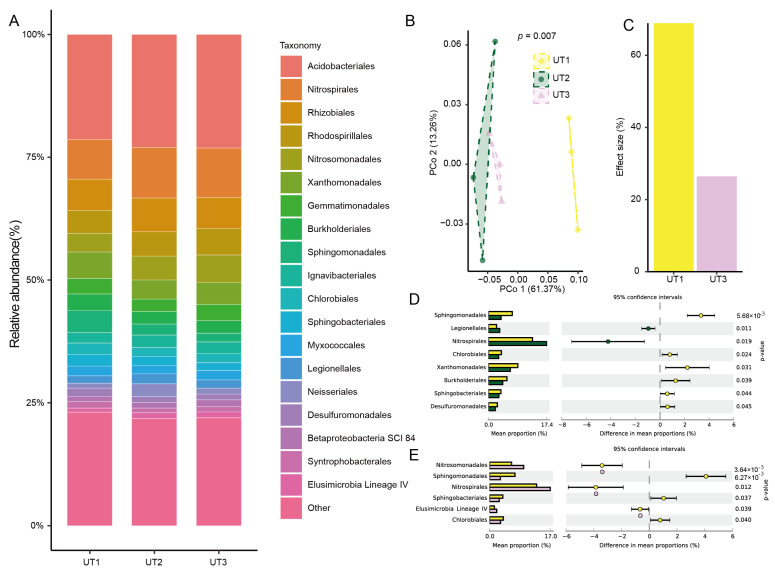
Effects of the sonication methods on bacterial community diversity and composition. (**A**) Relative abundance of bacterial communities treated with UT1, UT2, and UT3 at the order level. (**B**) The unconstrained PCoA based on Bray–Curtis distance showed bacterial community aggregation under the different treatments (*p* < 0.001, *p*-value calculated using PERMANOVA). (**C**) The extracted bacterial community from each method was compared to that of the UT2 method during PCoA followed by PERMANOVA, and the effect size of the extracted method was plotted in decreasing order. Differences analysis of top 10 orders among the UT1- and UT2-extracted microbiota (**D**) and the UT1- and UT3-extracted microbiota (**E**). The histogram shows the difference in bacteria abundance between the two groups. The dot and bar plots show the percentage of bacteria between the two groups in each sample. The difference in proportions between the groups is shown with 95% confidence intervals. Only *p*-value < 0.05 (Welch’s *t*-test) is shown. Each method was repeated three times. The abbreviations and different colors are the same as those shown in Figure 3.

**Figure 5 microorganisms-12-00403-f005:**
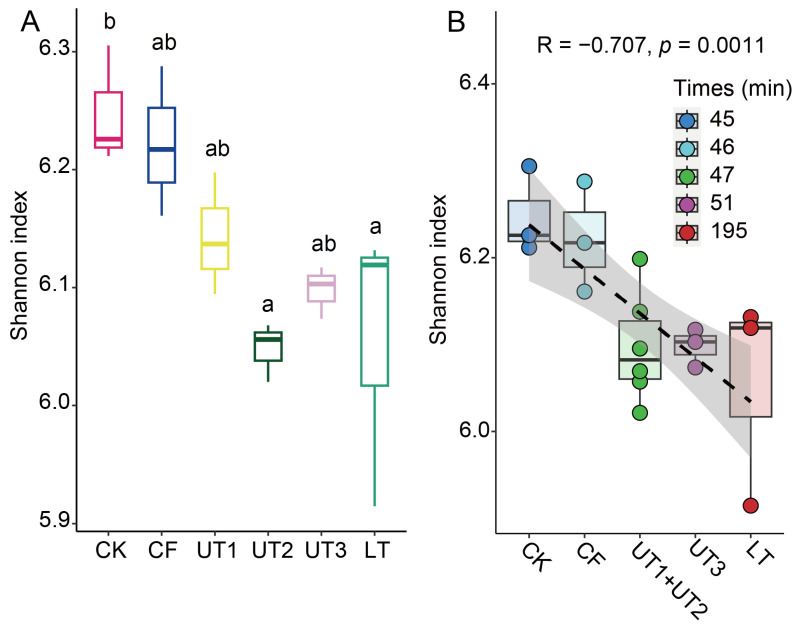
Comparison of α-diversity in the bacterial communities under different treatments. (**A**) Shannon index of bacterial communities under different treatments. Through one-way ANOVA compared with Tukey’s test, the same letters indicate no significant difference, and different letters indicate a significant difference (*p* < 0.001). (**B**) Correlation analysis between the processing time and Shannon index. The processing time of each extraction method was analyzed in increasing order (UT1 is the same as UT2). The dashed line represents a negative correlation between the processing time and Shannon index (R = −0.707, *p* = 0.0011). The scatter plot represents the individual sample data, and the top and bottom of the box represent the 75th and 25th percentiles, respectively. The line in the box represents the median. The upper and lower parts must extend from the upper and lower edges of the box, respectively, for data within a range of no more than 1.5 quartiles. The abbreviations are the same as those shown in Figure 1 and Figure 2.

**Table 1 microorganisms-12-00403-t001:** Details of the soil microbiota extraction methods.

Method Name	Oscillation (200 rpm)	Sonication (30 kHz)	Centrifugation (600 rpm, 4 °C)
CK	30 min	-	-
CF	30 min	-	1 min
UT1	30 min	2 min	-
LT	180 min	-	-

## Data Availability

The raw sequence data (16S rRNA gene fragment sequencing) generated in this study have been deposited in the Genome Sequence Archive of the BIG Data Center [48], Chinese Academy of Sciences, under accession number PRJCA021722 (https://ngdc.cncb.ac.cn/bioproject/browse/PRJCA021722, accessed on 1 January 2024).

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
