# Peer review of "Extraction Methods Determine the Quality of Soil Microbiota Acquisition"

_microorganisms, 2024, doi:10.3390/microorganisms12020403_

Round 1

Reviewer 1 Report

Comments and Suggestions for Authors

The manuscript "Extraction methods determine the quality of soil microbiota acquisition" is an interesting study presenting methods for the extraction of culturable soil microbiota. According to the content of the paper, I recommend the authors to change the title to emphasise that only culturable microbiota are analysed (as I understand it). Also, in line 73, it should be „as CF method” not “LT method”. Line 77 - based on the analyses... how do you know that sonication was the main factor? It would be better if you listed all tested methods in the table. Fig. 2 and 4. the figures are too small, it is difficult to read the content.

Reviewer 2 Report

Comments and Suggestions for Authors

I’ve found in the introduction several sentences which appear not in the correct section, may be more related to results or conclusion.

In the material and methods section, the extraction methods should be 4, but in the text the details about the method called “CF” are missed or not clearly explained.

It poor clear how bacteria colonies obtained in cultures were evaluated in their diversity, for example in terms of number of taxa recovered. In my opinion it could be useful to sequence by Sanger the colonies obtained to have the chance to also qualitatively compare the results in terms of diversity.

I do not understand why in the 16S microbiota diversity, the analysis was conducted up to order level and not at genus level. At this porpoise I found into the caption of Figure 2 the term “species” used improperly considering that the level of the analysis is “order”.

Comments on the Quality of English Language

Many parts of the text appear confusing and not very clear. It is very hard to follow the meaning.
